# T and Small Protrusion (TAP) Technique in Bifurcations: Coronary Artery Disease in Acute Myocardial Infarction Patients after COVID-19 Pneumonia

**DOI:** 10.3390/biomedicines11082255

**Published:** 2023-08-11

**Authors:** Marius Rus, Georgiana Carmen Filimon, Adriana Ioana Ardelean

**Affiliations:** 1Cardiology Clinic, Bihor County Emergency Clinical Hospital, 410167 Oradea, Romania; georgiana.cartis@gmail.com (G.C.F.); adriana_ardelean@uoradea.ro (A.I.A.); 2Faculty of Medicine and Pharmacy, University of Oradea, 410610 Oradea, Romania

**Keywords:** bifurcation lesions, DK-Crush, TAP, STEMI, coronary artery disease

## Abstract

Ischemic coronary artery disease in all its forms remains the main cause of death worldwide. Coronary artery bifurcation lesions are a challenge because of their complexity and possible complications. The goal of treating bifurcation lesions is the optimal revascularization of the main vessel without compromising the side branch. Although the study of bifurcation stenting aims to keep the side branch viable, the outcomes regarding major acute cardiovascular events and survivability are related to the optimal treatment of the main vessel. There are many trials that have tried to evaluate the best technique to use with respect to bifurcation lesions, and early studies support provisional stenting as the election treatment. More recent trials highlighted the superior outcomes of the double kissing crush technique used on unprotected distal left main bifurcation lesions. In patients with acute myocardial infarction, two-stent techniques were avoided because of the prolonged procedural time in unstable patients, with high risks of complications. We present the case of a 53-year-old woman with multiple cardiovascular risk factors (dyslipidemia, hypertension, active cancer, post-COVID-19 state) and acute antero-lateral myocardial infarction who underwent primary coronary intervention with the use of the TAP technique for stenting the bifurcation culprit coronary lesion (left anterior descendent artery and first diagonal artery).

## 1. Introduction

Coronary artery ischemic disease is the major cause of death worldwide. Atherosclerosis is considered the major risk factor for myocardial ischemia, and it is a complex process that involves a series of events like lipid accumulation in the vessel endothelium, the activation of inflammatory factors (macrophages, interleukins, and cytokines), fibrous elements, and later, calcification [1]. 

Coronary bifurcation lesions represent more than 20% of coronary artery stenosis undergoing angioplasty [2]. A bifurcation lesion is defined as a lesion that occurs adjacent to a division of a major epicardial vessel. There were several attempts to classify these lesions, and currently, there are six used classifications. However, none of them provide strict therapeutic conduct for bifurcation lesions. The most important intra-procedural classification divides bifurcation lesions into true, which is when both the main vessel (MV) and the side branch (SB) present ostial stenotic lesions, and nontrue, which includes all different stenotic involvements of a bifurcation. The Medina classification is the most commonly used of the six classifications (Figure 1) [3]. The distal significant stenosis of the left main coronary artery (LMCA) is considered a nontrue bifurcation. 

The main question remains with respect to whether a bifurcation lesion needs to be treated with the provisional stenting of the main vessel or whether a two-stent technique is required from the start. In most cases, the provisional stenting of the MV is the conduit of choice, with a protective “keep it open” guide wire placed in the SB. However, stenting the side branch in a true bifurcation takes the following into consideration: if the size of the side branch is under two-millimeter (mm) diameter, which usually means that the vessel does not need to be preserved, or if the supplied myocardial territory is small. Usually, the strategy depends on the risk of SB occlusion. The high-risk factors for SB occlusion are as follows: ostial localization of the lesion, severity of stenosis, the angle between MV and SB, and the morphology of the stenosis [4]. After provisional MV treatment, the SB is considered to be damaged in the case of severe dissection or the emergence of stenosis that is greater than 50%. This requires changing the strategy into a two-stent technique. Bifurcation lesions were highly studied because of the vast territory at risk, and with the development of new-generation drug-eluting stents, more two-stent techniques became available. Mohamed MO et al. considered proximal optimization (POT) to be essential as a final step in two-stent techniques, as a better outcome was observed in patients with bifurcation coronary artery disease when POT was made [5].

### 1.1. Anatomy Changes of Bifurcation Lesions 

The main causes of ostial SB compromise after stenting the main vessel are the shift of the carina or the plaque shift. Previously, the theory regarding the aggravation of the ostial lesion of the side branch was the shifting of atherosclerotic plaque from the MV to the SB [6]. Plaque shifting is defined as an increase in plaque volume in the ostial side branch after the provisional stenting of MV. The plaque shift phenomenon is known as the “snowplow effect”, and it occurs more frequently after stent deployment than after balloon angioplasty [2]. Intravascular imaging has shown that the carina walls are usually free of atherosclerotic disease due to high blood flow, and recent studies suggest that carina shift is the major mechanism of SB intraprocedural occlusion [7,8]. Carina shift is defined as a vessel volume decrease after MV stenting, and it is associated with the functional compromise of the vessel, which can be objectified via FFR [9]. Anatomy changes in the atherosclerotic plague are a strong determination of restenosis, and because of this, the development of atherectomy techniques increased. However, studies have shown no clinical or angiographic benefit, resulting in their limited usage in niche cases [10]. 

### 1.2. Bifurcation Two-Stent Techniques

Although the general consensus is to treat coronary artery bifurcation lesions with provisional stenting, with the possibility of changing to a two-stent technique if there is significant side branch damage, there are particular cases where a special bifurcation technique is needed from the start. There are multiple two-stent techniques, and the choice between them is made considering the angle between MV and SB and the severity, anatomy, and morphology of vessel stenosis. 

#### 1.2.1. T-Stenting and TAP (T and Small Protrusion) Technique

T-stenting and TAP techniques are mostly used in bifurcations in which the angle between the two branches is close to 90°, especially if the MV is significantly larger than the SB. These are easier techniques than Culotte or DK crush, with fewer steps required. In the T-stenting technique, the first stent is placed at the origin of the side branch so that the entire ostium is covered. The second stent is deployed in the main vessel, followed by a side branch and main vessel simultaneous kissing balloon inflation (KBI). The main problem remains the possibility of incomplete coverage of SB ostium, which can lead to artery restenosis. The T and small protrusion technique is a modified T-stenting technique where the first stent is deployed in the MV. The stent from the SB is intentionally positioned with a small (1–2 mm) protrusion in the MV and is deployed simultaneously with a balloon positioned in the MV. The TAP technique provides full coverage of the SB ostium, but the drawback of TAP remains in the creation of a single-layer stent neocarina [11]. 

#### 1.2.2. Culotte Technique

The Culotte technique is mostly used in bifurcation lesions with an angle of less than 70° between SB and MV, and it is best suited in cases with similar vessel sizes [4]. This technique provides the best coverage of the carina, with the secondary drawback of an excess of metal at the proximal end and the carina. After the preparation of both branches, a stent is deployed in the most angulated vessel (more frequently, the side branch). The following step is to rewire the non-stented vessel (more often, the main vessel) and perform balloon dilatation. A stent is deployed in the MV, and a final KBI is carried out. 

#### 1.2.3. Double-Kissing Crush (DK Crush)

DK crush is a more difficult technique; it requires multiple steps, each of them with potential difficulties and complications. The crush techniques were introduced to minimize the stent overlap between SB and MV [12]. The main advantage of DK crush is that it can obtain an immediate patency of both branches, and its main disadvantage is the fact that is a laborious technique that requires multiple stent rewiring. DK crush is the most studied of the two-stent techniques, and trials have shown a 5% target lesion failure rate at 12 months versus 10% in the provisional group in stenting left main stenosis [13]. 

#### 1.2.4. Simultaneous Kissing Stents (SKS)/V-Stenting

This is the most frequently used two-stent technique in emergency stenting. The general concern in this case is that a stent neocarina is formed at the center of the proximal MV, which is unlikely to fully endothelialize. Also, this new stent carina introduces further wiring difficulties if a second revascularization is needed [2]. The main advantage is that it never loses the aces of any of the two branches, with no need to recross any stent. SKS is mostly used in proximal lesions, and it leads to the formation of a double-barrel new proximal carina. V-stenting is preferred when the lesions are distal to the bifurcation (Medina 0.1.1). 

## 2. Case Presentation

Patient S.D., female, 53 years old, known to have cervix squamous carcinoma (for which she underwent chemotherapy and radiotherapy) and a recent history of COVID-19 mild pneumonia, presented to the Emergency Department while complaining of retrosternal chest pain with constrictive character and a sudden onset five hours from presentation. 

Objective: diaphoresis; obesity (body mass index = 35 kg/m^2^); BP = 103/80 mmHG; RR = 16/min; VR = 70/min; oxygen peripheral saturation = 92% environmental air.

Electrocardiogram at presentation (Figure 2): sinus rhythm, VR = 72/min, ST segment elevation max 5 mm in DI, aVL, V2–V6, and ST segment depression 4 mm in DIII, aVF.

The patient underwent emergency coronarography, which revealed proximal thrombotic left descending artery (LAD) occlusion and normal right coronary artery (RCA), left coronary main (LM), and circumflex coronary artery (CXA). 

Before the procedure, the patient received a 180 mg loading dose of Ticagrelor, then continued with 90 mg of b.i.d.; 300 mg loading dose of Aspirin, then continued with 75 mg/day; and 100 U/kg of unfractionated heparin was used during the PCI procedure.

Primary percutaneous coronary artery angioplasty (PTCA) was needed. An EBU catheter was positioned in the left coronary artery ostium. The LAD lesion was crossed with extra-floppy wire, and as a first step, balloon pre-dilatation was carried out with a semi-compliant 2.0/20 mm balloon that was expanded at 4 atm. A 4.0/26 mm drug-eluting stent (DES) was deployed to treat the proximal LAD lesion. The first diagonal artery (DI) was wired with a floppy wire, and we decided to change the procedure using the two-stent technique because of significant DI ostial stenosis. The ostial DI lesion was predilated with a 2.0/20 mm semi-compliant balloon that was expanded at 12 atm. A 3.5/19 mm DES stent was positioned in DI with a minimal protrusion in the LAD; then, a non-compliant 3.5/10 mm stent was positioned in LAD, which covered the DI ostium. The stent was deployed by expansion at 12 atm simultaneously with the inflation of the LAD balloon at 12 atm (TAP technique). Final kissing balloon inflation was carried out. In the proximal LAD, we used a non-compliant 4.5/10 mm balloon for the proximal optimization technique (POT) (Figure 3).

Post revascularization EKG: sinus rhythm, VR = 80/min, ST elevation 2 mm in V2–V3, DI, aVL, inverted T wave DI, aVL, V1–V3 (Figure 4). 

Two-dimensional and three-dimensional transthoracic echocardiography revealed an impaired left ventricle ejection fraction (LVEF) of 38%, assessed using the Simpson formula, left ventricle aneurismal akinetic apex, with an apical LV thrombus, mild mitral regurgitation, aortic and tricuspid normal valve, and normal right ventricle dimensions and function (Figure 5). 

The laboratory blood sample (Table 1) revealed increased myocardial enzymes, leukocytosis, hyperglycemia, and normal renal function. 

Pharmacological post-procedure treatment included high-dose atorvastatin, proton-pump inhibitors, dual anti-platelet therapy, and diuretics. After the assessment of the apical LV thrombus, anticoagulant therapy was needed, with the initiation of fractioned heparin (enoxaparin 0.1 mg/kg b.i.d.) and changing ticagrelor with 300 mg of clopidogrel as loading dose, which continued at 75 mg/day (according to ESC guidelines).

Diagnosis: ST segment elevation acute myocardial infarction (STEMI) was observed in antero-lateral territory. Primary coronary intervention (PCI) with the implantation of two stents (LAD and DI) was carried out. Left ventricle apical aneurism. Left ventricle apical thrombus. Mild mitral regurgitation. Cervix squamous carcinoma (previously treated with chemotherapy and radiotherapy). Post-COVID pneumonia status.

For this case report, we chose a patient who suffers from two major pathologies that favor clotting formation (SARS-CoV-2 infection and cancer). Data collection was carried out from the in-hospital care of the patient using local hospital electronic systems. Data collection and analysis were carried out by all authors, with equal contributions.

## 3. Discussions

The interventional treatment of coronary artery bifurcation lesions is associated with more frequent short- and long-term complications due to complex techniques, longer intra-procedural time, and a greater risk of restenosis. A complete revascularization of both MV and SB should provide a better result, but multiple studies showed that complex two-stent techniques have a worser outcome because of the prolonged procedural time, multiple steps, and possible complications during every step. The European Bifurcation Club suggests that provisional stenting should be the first choice for the interventional treatment of bifurcation coronary artery, leaving two-stent techniques for complex lesions with an important side branch that supplies large myocardial territory [14]. The Bifurcation Academic Research Consortium (Bif-ARC) developed a series of criteria regarding the side branch for a lesion to be considered complex: vessel size, the importance of the SB (distribution and territory), the extent of calcification, thrombus, and lesion geometry [15].

Because of the large myocardial territory at risk of ischemia, multiple trials were carried out to establish the best method for treating bifurcation lesions. In left main stenosis, coronary artery bypass grafting (CAGB) was preferred, especially in diabetic patients. The NOBLE trial compared 5-year outcomes in patients treated with PCI or CABG for left main lesions, the primary endpoint being major acute cardiovascular events (MACEs). The results have shown a better outcome for the CABG group because patients treated with PCI required revascularization for restenosis [16]. However, the EXCES study revealed that unprotected left main stenosis with a Syntax score under 32 points is more beneficial for PCI than for CABG. One of the first trials regarding coronary bifurcations was the NORDIC bifurcation I study, which compared provisional stenting with two-stent techniques. The results showed a similar MACE (death, MI, TLR, and stent restenosis) in the two groups, but a longer procedural time associated with higher rates of increased biomarkers for the two-stent techniques. DK crush is the most studied two-stent technique, and a series of trials compared it with all other stenting methods. The DK Crush I trial studied the parallel between classic crush technique with DK crush, and the results showed lower MACE rates at 8 months, exhibiting an easier achievement of 100% final kissing balloon inflation for patients treated with DK crush [17]. The DK crush III trial studied unprotected distal left main stenosis (Medina 1.1.1 or Medina 0.1.1) and carried out a comparison with respect to outcomes regarding MACE between the DK crush technique and Culotte, resulting in higher MACE rates following the Culotte technique. The reason was higher target vessel revascularization [18]. 

Provisional stenting is the most simple technique for bifurcation lesions, and it can be converted into TAP or Culotte if needed and remains the more indicated and used treatment method. However, for left main distal stenosis, the DK Crush V trial demonstrated lower rates of MACE for patients who underwent the DK crush technique than those with provisional stenting [19]. In our patient’s case, we chose initial provisional stenting, and after the evaluation of SB importance, we decided to convert our technique into TAP. 

After observing coronary anatomy, we assessed the TIMI risk score as equal to 1 point, which indicates a low rate of 30-day mortality (1.6%). The RISK-PCI score was developed to evaluate 30-day major acute cardiovascular events (MACE) in patients presenting STEMI and who underwent primary PCI [20]. In our case, the RISK-PCI score was 4.5 points, which corresponds to an intermediate risk for MACE (5–8.8%). The SYNTAX score was designed to evaluate post-procedural complication risks, considering the anatomy and atherosclerotic burden of coronary arteries. The score is graded from 0 (simple lesions) to over 60 (very complex lesions). Studies have shown that a patient with a SYNTAX score of less than 33 has equal outcomes after revascularization regarding the MACE for both PCI and CABG (coronary artery bypass grafting) [21]. In our patient’s case, the SYNTAX I score was 25.5 points, and SYNTAX score II was 38.2, so we decided that PCI was the most suitable technique. 

Multivessel coronary stenting and acute coronary syndrome are two of the major predictive risk factors for post-PCI complications. The most important complications are artery dissection, perforation, sudden vessel collapse, and no-reflow phenomenon. Microvascular occlusion is characterized by the absence of blood flow (“no-reflow”) in the coronary artery after stent insertion. This is caused by myocardial microvascular injury and dysfunction, so blood flow cannot pass through the myocardial capillary [22]. The pathophysiology of microvascular occlusion begins with a vulnerable base status of the capillary endothelium, such as endothelial dysfunction from diabetes mellitus, and it is favored by the distal embolization of the thrombus after restoring coronary flow and by intravascular inflammatory factors [23].

Coronary artery disease is a frequent comorbidity in cancer patients explained by cardiotoxic chemotherapy and thoracic radiotherapy and by the hypercoagulable status that predisposes individuals to thrombotic events. Patients with STEMI and active cancer have significantly higher all-cause mortality at 1 year [24,25].

Viral infections, especially involving the respiratory tract, are a possible risk factor for acute myocardial infarction (AMI) due to the hypercoagulable state of infections [26,27]. Although asymptomatic patients were assumed to have a lower viral burden, recent data showed that asymptomatic SARS-CoV-2-positive patients presenting with STEMI have higher thrombus viral loads, higher thrombus dimensions, and poorer myocardial blush grades assessed by coronarography [28]. COVID-19 survivors had an increased risk for AMI, and their incidence remains higher than in the control group until 8.5 months of follow-up. The risk of AMI returns to the baseline within a few months after the resolution of the infection, and it is more pronounced among patients with more severe disease [29,30,31]. Further studies discovered that COVID-19-infected patients and recovered patients seem to exhibit a remaining hypercoagulable state. 

The populations with a higher risk of infection with SARS-CoV-2 and worse prognosis are patients with underlying heart disease (coronary artery disease, heart failure, and hypertension), pulmonary disease (asthma and cystic fibrosis), obesity, patients with diabetes mellitus, and immunodepression. Patients with reduced response to vaccination (mostly because of therapeutical immunosuppression or immune disease) have a higher risk of severe COVID-19 pneumonia and worse outcomes. Hypertension is still the most common comorbidity in COVID-19 patients, but the outcome of these patients is possibly linked with blood type. A recent prospective study showed that patients with the O blood type infected with SARS-CoV-2 that present hypertension, have significantly lower values of pro-thrombotic indexes and lower rates of cardiac injury and deaths compared to non-O patients [32].

## 4. Conclusions

Complex bifurcations coronary artery lesions have a worse intermediate and long-term outcome than simple lesions. In STEMI patients, the provisional stenting of the culprit lesion is preferred. However, in some cases, when the side branch lesion increases the ischemic territory, a two-stent technique may be used upfront. Respiratory tract infections, including SARS-CoV-2 infections, induce a hypercoagulable status, increasing the risk for acute coronary syndromes.

## Figures and Tables

**Figure 1 biomedicines-11-02255-f001:**
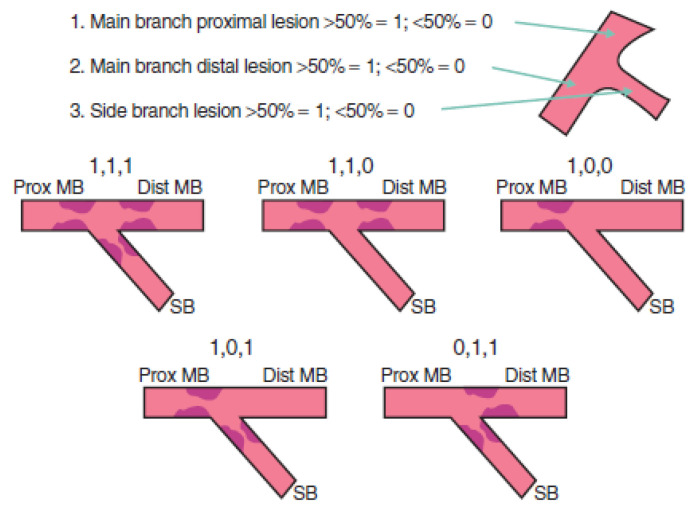
Medina classification [3].

**Figure 2 biomedicines-11-02255-f002:**
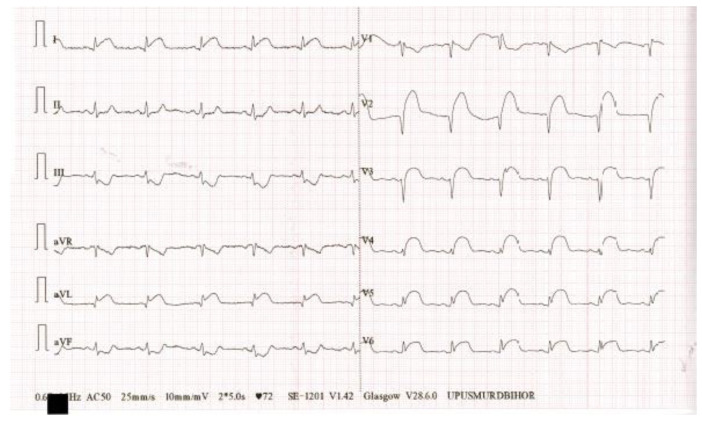
EKG at presentation.

**Figure 3 biomedicines-11-02255-f003:**
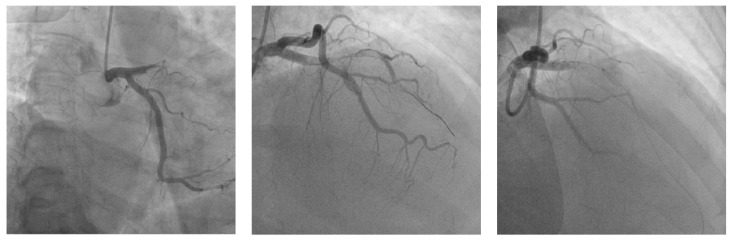
Revascularization of LAD and DI arteries using the TAP technique.

**Figure 4 biomedicines-11-02255-f004:**
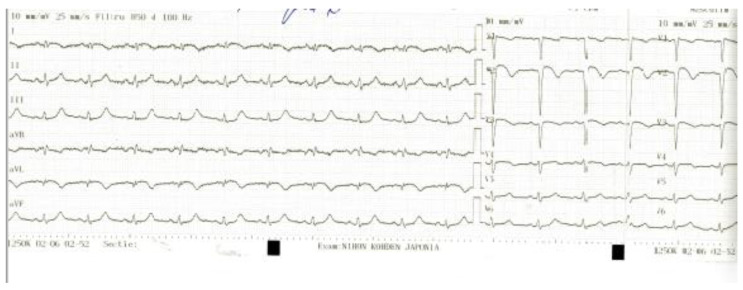
Post-revascularization EKG.

**Figure 5 biomedicines-11-02255-f005:**
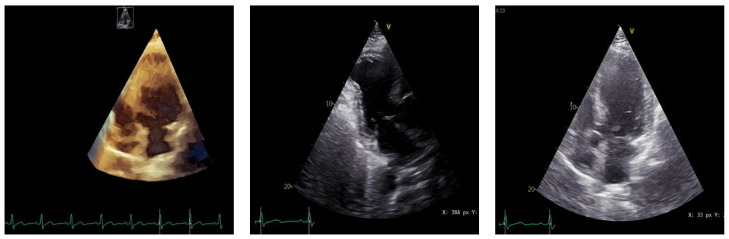
Transthoracic echocardiography.

**Table 1 biomedicines-11-02255-t001:** Laboratory blood sample.

Assay	Value	Normal Rage
WBC	14.2 × 10^3^	4.00–10.00 × 10^3^
GLU	131 mg/dL	74–100 mg/dL
GGT	38 U/L	9–36 U/L
CREA	0.78 mg/dL	0.57–1.11 mg/dL
CK-MB	592 U/L	0.00–24.0 U/L
HS-TROP	10,003.1 pg/mL	0.00–15.6 pg/mL

## Data Availability

The data support for this case report article are not public do to patients privacy.

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
