# Peer review of "T and Small Protrusion (TAP) Technique in Bifurcations: Coronary Artery Disease in Acute Myocardial Infarction Patients after COVID-19 Pneumonia"

_biomedicines, 2023, doi:10.3390/biomedicines11082255_

Round 1

Reviewer 1 Report

The authors submited a case report of coronary artery bifurcation lesions management. The case report is well written and composes of historical aspect of the issue, clear description of the case along with discussion around of desicion-making and choice of the treatment. The presentation sound well, while I would like recommend the following.

The authors should give more information regarding pre- and post-PCI treatment and evaluate an impact of coronary anatomy on a risk of post-PCI complication(s) icluding microvascular obstruction etc.

Author Response

Point 1 - The authors should give more information regarding pre- and post-PCI treatment and evaluate an impact of coronary anatomy on a risk of post-PCI complication(s) including microvascular obstruction etc.

Response to Reviewer 1 Point 1 – We included in our Case Report information about pre- and post-PCI pharmacological treatment. We also included the assessment of risk stratification for 30-days mortality and MACE for this patient. We added the evaluation of risk of post-PCI complications, according to the patient’s coronary anatomy. Thank you!

Reviewer 2 Report

An interesting and informative manuscript that has clinical merit.  However, there editing issues that the authors should consider and address.  Teh following are suggestions/comments regarding those issues.  Line 72, "However, studies have shown no clinical ...".  Line 79, "... the angle between MV and SB, ...".  Lines 92 & 93, "... the drawback of TAP remains in the creation ...".  Line 108, "... two-stent techniques in which trials have shown a 5% target lesion failure rate at 12 months ...".  Lines 112 & 113, "... center of the proximal MV, which is ...".  Line 122, "... presents to the Emergency Department complaining of retrosternal ...".  Line 123, "... constrictive character of sudden onset, five ...".  Line 143, "... inflation is made.  In the proximal LAD we ...".  Line 144, "... balloon for a proximal optimization ...".  Line 173, "... of both MV and SB should provide ...".  Line 183, "Because of the large myocardial ...".  Line 189, "The results have shown a better outcome for ...".  Line 196, "... biomarkers for the two-stents technique.  DK Crush ...".  Line 205, "... Culotte if need be and remains the ...".  Line 215, "... less than 1 year, if the platelet count is ...".  Line 220, "higher than in the control group until 8.5 months of follow-up."  Line 229, "term outcome than simple lesions."

A well written manuscript.

Author Response

Point 1 - An interesting and informative manuscript that has clinical merit.  However, there editing issues that the authors should consider and address.  The following are suggestions/comments regarding those issues.

Response to Reviewer 2 Point 1 – We corrected the editing and spelling issues found in the manuscript, according to these suggestions. Thank you!